# Bridging performance gap between minimal and maximal SVM models

**Ondrej Šuch**                                        *ondrejs@savbb.sk*
*Mathematical institute of Slovak Academy of Sciences*
*Ďumbierska 1*
*Banská Bystrica, 974 01, Slovakia*

**René Fabricius**                              *rene.fabricius@fri.uniza.sk*
*Faculty of Management and Informatics*
*Žilinská Univerzita v Žiline*
*Univerzitná 8215/1 Žilina, 010 26, Slovakia*

**Reviewed on OpenReview:** *https://openreview.net/forum?id=SM1BkjGePI*

## Abstract

Multi-class support vector machine (SVM) models are typically built using all possible pairs of binary SVM in a one-against-one fashion. This requires too much computation for datasets with hundreds or thousands of classes, which motivates the search for multi-class models that do not use all pairwise SVM. Our models correspond to the choice of the model graph, whose vertices correspond to classes and edges represent which pairwise SVMs are trained. We conduct experiments to uncover metrical and topological properties that impact the accuracy of a multi-class SVM model. Based on their results we propose a way to construct intermediate multi-class SVM models. The key insight is that for model graphs of diameter two, we can estimate missing pairwise probabilities from the known ones thus transforming the computation of posteriors to the usual complete (maximal) case. Our proposed algorithm allows one to reduce computational effort by 50-80% while keeping accuracy near, or even above that of a softmax classifier. In our work we use convolutional data sets, which have multiple advantages for benchmarking multi-class SVM models.

## 1 Introduction

With ever-increasing computational capacity requirements needed for training deep neural networks, transfer learning has emerged as a cost-effective alternative for constructing classification models for computer vision and related application areas (Bengio, 2012; Niu et al., 2020). One method and the one that motivates our work, to prepare a transfer learning model is to start with a neural network pre-trained on a computer vision dataset $D$ such as Imagenet (Huh et al., 2016), extract activations of neurons from the penultimate layer on the target dataset $D'$ and use these features to train a classifier model $M'$ for $D'$.

If it is reasonable to assume that the boundaries among classes are linear, then one may opt to build $M'$ by training a multinomial regression (softmax) model. The main potential drawback to softmax classifiers is that location of class boundaries may be skewed by the presence of outliers in the dataset. This drawback could be addressed by instead choosing a support vector machine model (SVM) for building classifier $M'$. For SVM the presence of outliers does not affect the learned boundary since outliers do not contribute to the loss function of SVM (Cortes & Vapnik, 1995). Moreover, while vanilla SVM learns linear class boundaries, more general kernel SVM is capable of learning a diversity of nonlinear class boundaries.

The underlying geometric concept of SVM, the division of space by a hyperplane into two halfspaces, is a natural fit for two-class classification problems. To apply SVM to multi-class classification problems one usually constructs an ensemble of SVM for multiple two-class problems and classifies by combining the

predictions of the ensemble's elements. For smaller datasets with a few tens of classes, SVM is a suitable alternative to softmax for building $M'$ (see e.g. Maitra et al. (2015); Raghu et al. (2020)).

It is increasingly common to have datasets with hundreds or thousands of different classes. For such datasets, building an SVM model using common SVM software libraries may become prohibitively expensive in terms of computational cost. The focus of our work is to investigate the possibility to reduce the computational cost to build an SVM model for a dataset where the number of classes $C$ is large (say $C \geq 50$). Although motivated by transfer learning, we are seeking a general method applicable to datasets unrelated to neural networks, and more general SVM kernels, notably the popular radial (RBF) kernel. Also, we aim to build probabilistic models which are more flexible in applications, and were shown to perform better than voting ensembles (Wu et al., 2004; Duan & Keerthi, 2005).

Previous works successfully investigated the more specialized problem of coopting SVM with a neural network (Tang, 2013; Passricha & Aggarwal, 2019). Thousand class Imagenet classification using an SVM modification on 2048 features from a convolutional network was achieved by Do (2021). Speeding up of SVM training by finding an approximate solution was proposed by Cao & Boley (2006). For linear kernels, the work of Fan et al. (2008) specifically aims for large datasets SVM classification.

## 1.1 Complexity of building SVM models

The computational cost of multiclass SVM for larger datasets is driven by two factors. The first is the number of individual SVM problems that need to be solved. For some approaches, the number of required models grows quadratically with the number of classes. The second factor is the cost to train each SVM. The training time of binary SVM scales superlinearly with the number of training samples (Rifkin & Klautau, 2004). Therefore it may well be beneficial to train a larger number of binary SVM for smaller datasets than to train a smaller number of binary SVM for larger datasets. Let us briefly examine this tradeoff by comparing the complexity of four popular approaches implemented in libraries LIBSVM (Chang & Lin, 2011), scikit-learn (Pedregosa et al., 2011), LIBLINEAR (Fan et al., 2008), and cvx-opt (Andersen et al., 2020). In the discussion below we will use the estimated time complexity of training a single SVM with the radial kernel as reported in Table 2.

A very common approach is to train SVM classifiers for all pairs of classes, which entails building $C(C-1)/2$ models. This approach is called one-vs-one (OVO), or sometimes all-vs-all (AVA). This yields an ensemble of classifiers, and proposed methods to combine its elements into a final classification include voting, building Directed Acyclic Graph (Hsu & Lin, 2002), or pairwise coupling (Chang & Lin, 2011; Hastie & Tibshirani, 1998).

Another common approach is to train SVM classifiers separating each class from the rest of the training dataset, which requires training $C$ SVM classifiers. This approach is called one-vs-all (OVA), or one-vs-rest. It yields models of similar accuracy to OVO (Rifkin & Klautau, 2004; Galar et al., 2011). For the 50 class subset of Imagenet studied in this paper, we found that the training complexity of each SVM for radial kernel may be approximated as $\sim C^{1.5}$, which makes this method asymptotically slower (complexity $\sim C^{2.5}$) than the one-vs-one model, which has complexity $\sim C^2$. A similar conclusion was reported by Fürnkranz (2002).

A method by Crammer and Singer (Crammer & Singer, 2001) modifies SVM to directly solve the multiclass problem without resorting to an ensemble. Documentation of the scikit-learn library notes however "while crammer-singer is interesting from a theoretical perspective as it is consistent, it is seldom used in practice as it rarely leads to better accuracy and is more expensive to compute [than OVA]" (scikit-learn, 2023).

Finally, let us mention error-correcting output codes (ECOC) which is a method inspired by coding theory (Dietterich & Bakiri, 1994). It chooses a fixed-length code of length $L$ and assigns a codeword to each class. A binary classifier is then trained for each bit, resulting in the need to train $L$ classifiers. ECOC models encompass a wide range of models depending on the code chosen, including the OVA model for $L = C$. We will analyze one variant of ECOC for the code achieving the lower bound on $L$ which is $\lceil \log_2 C \rceil$. Suppose $C = 2^L$. A decision by a single ECOC SVM provides at most one bit of information. It provides exactly one bit of information, if and only if the number of classes for both negative and positive samples is $C/2$. For the 50 class Imagenet dataset, we model that the complexity of building such an SVM model is $\sim C^{2.1}$,

which translates to the overall complexity of $\sim C^{2.1}\log(C)$, thus being approximately equal to the quadratic cost of the OVO model. Given that training SVM for this ECOC model involves many more samples than the OVO model, on computers or graphic cards with limited memory, the OVO method will likely be much faster.

Despite OVO being asymptotically the fastest algorithm, its quadratic complexity makes it impractical for larger values of $C$. To make it faster, we propose to construct a multi-class SVM model by training only a subset of all possible pairwise SVMs. We will refer to the subset as the *model graph* with vertices corresponding to the classes.

### 1.2 Minimal, maximal, and intermediate models

The starting point of our investigation is the observation (see Section 2.5), that an SVM model can be built by training just $C-1$ binary SVMs, each trained in a one-vs-one fashion, and then converted to a probabilistic model using Platt's method (Platt et al., 1999). We refer to these models as *minimal models*. There are many minimal models since minimal models are in one-to-one correspondence with spanning trees on $C$ vertices. Minimal models are faster to train than other multi-class SVM algorithms. The training time advantage comes at a cost of decreased accuracy of minimal models, as we show in Section 3.1. As a benchmark, we compare the minimal models with a commonly used coupling methodology that requires all $\binom{C}{2}$ pairwise binary SVMs trained in a one-vs-one fashion. We refer to the latter as *maximal models*.

Guided by the Pareto principle, we investigate the possibility to increase the accuracy of minimal models by incrementally adding a moderate amount of additional binary SVMs to capture most of the accuracy advantage of the maximal model. We will answer the following key questions needed to formulate a viable algorithm to construct such *intermediate models*:

Q1. how to choose the initial set of $C-1$ binary SVMs,

Q2. how to make a multi-class prediction using a non-complete set of probabilistic binary SVMs trained in a one-vs-one fashion,

Q3. how to choose which binary SVM to incrementally train and add to the multi-class model.

## 2 Experimental methodology

### 2.1 Software used for experiments and analysis

For computations of SVM, we use R package e1071 (R Core Team, 2021; Meyer et al., 2020). For data processing, we use tidyverse (Wickham et al., 2019) and for visualizations package ggplot2 (Wickham, 2016). The source repository can be found on `https://github.com/ondrej-such/svm3`.

### 2.2 Datasets

In this work, we use datasets for classifying images based on input features to the final softmax layer of a convolutional neural network. This class of problems has several compelling attributes for benchmarking SVM algorithms, namely

- the set of classes is well defined (e.g. in comparison with much less clear phoneme classes in speech processing),

- benchmark image datasets have been thoroughly examined for the correctness of annotation; moreover, the correctness of annotation can be easily assessed by a layperson,

- convolutional neural networks are trained with the softmax layer to high classification accuracy, which suggests that inter-class boundaries are close to linear,

| Dataset name | Classes | # samples when training a neural network (per class) | # samples when training SVM (per class) | # samples in the testing dataset (per class) | Image resolution | # of SVM features |
|---|---|---|---|---|---|---|
| CIFAR-10 | 10 | 6000 | 10000 | 1000 | 32x32 | 512 |
| Imagenette | 10 | $\approx 1000$ | 10000 | 50 | variable | 512 |
| Imagewoof | 10 | $\approx 1000$ | 10000 | 50 | variable | 512 |
| Imagenet-50 | 50 | $\approx 1000$ | 10000 | 50 | variable | 512 |

Table 1: Summary of datasets used in the experiments. Train samples for SVM include augmented data samples.

- digital image classification has widespread applications, and any potential improvements would be quite valuable.

An overview of the datasets used throughout the paper is summarized in Table 1. The number of classes was selected to balance the requirements of being able to run multiple experiments to get error bounds and to represent a large-class SVM problem while being within our computational capacity.

In sections 3 and 4 we have used three datasets each having ten classes: CIFAR-10 (Krizhevsky et al., 2009), Imagenette, and Imagewoof (Howard, 2022), the latter two being subsets of the well-known Imagenet 2012 dataset (Russakovsky et al., 2015). In section 4 we have used a subset of Imagenet created by choosing 50 classes at random among the thousand classes in ImageNet 2012. We refer to the set as Imagenet-50. Its classes are listed in the subdirectory imagenet_subsets of the code repository. We have tried to model the time complexity $T$ of fitting a single radial SVM depending on the number of classes $C$ encompassing two binary classes. We use the simple linear regression model

$$\log(T) = \alpha + \beta \log(C) + \epsilon. \tag{1}$$

Values of $C$ were taken at random between 10 and 21, and we tried two kinds of groupings of $C$ classes into positive and negative samples - OVA and ECOC. For the latter, only even values of $C$ were considered. The results are summarized in Table 2.

| Grouping | # of positive classes | # of negative classes | $\beta$ | Std. error of $\beta$ | Adjusted $R^2$ |
|---|---|---|---|---|---|
| OVA | 1 | $C-1$ | 1.5 | 0.1 | 0.94 |
| ECOC | $C/2$ | $C/2$ | 2.1 | 0.08 | 0.99 |

Table 2: Parameters of linear models (1) fitted to timing data.

## 2.3 Neural networks

For CIFAR-10 we have used a custom architecture by David C. Page, which can be quickly trained to 94% accuracy (Page, 2019). For Imagenette and Imagewoof datasets, we used the Resnet-18 network (He et al., 2016). This architecture was designed to solve the Imagenet classification problem, which has 1000 classes. We have trained 20 instances of each architecture on CIFAR-10 and Imagenet respectively. From the networks, we extracted convolutional features in experiments in this paper (the exception being the experiment in section 3.5).

## 2.4 Pairwise models for convolutional data sets

Given a convolutional neural network trained, we proceed by extracting the activations of the penultimate layer on a chosen dataset. All network architectures in this paper use 512 neurons in this layer. These 512

activations constitute features of our convolutional data sets. Each training matrix had 10000 entries per class, which included all training samples and also augmented data samples.

The next step is the computation of pairwise likelihoods for each pair of classes. One could do this by simply training an SVM model and applying Platt's method to estimate the likelihoods on the test data. However, we employ a slight modification akin to the procedure used in the foundational study of Wu et al. (2004). Namely, we divide 10000 samples into four equal-sized subsets and train an SVM for each subset. We apply Platt's method to derive pairwise likelihoods for each of the four models and then average resulting probabilities using geometric mean. An immediate advantage is the reduction of training time, since the training time of SVM scales superlinearly with training size (Abdiansah & Wardoyo, 2015). Moreover, by averaging, one may expect to arrive at more precise likelihoods.

We used default parameters for the svm function from the e1071 library. In particular, data were scaled, and we used default kernel and cost parameters.

## 2.5 Probabilistic prediction for minimal and maximal models

Consider a general multi-class classification problem of assigning to a given sample a probabilistic distribution $\mathbf{p} = (p_1, \ldots, p_C)$ among $C$ classes. For simplicity, we will index the $C$ classes by integers from one to $C$.

Suppose pairwise SVM models are trained for some subset $E$ of edges of the complete graph on $C$ vertices $1, 2, \ldots, C$. We assume that for any edge $(i, j)$ in $E$ the corresponding model $M_{i,j}$ gives a probabilistic prediction i.e. predicts that with probability $r_{ij}$ the sample belongs to class $i$, and with probability $1 - r_{ij}$ it belongs to the class $j$. We note that throughout this paper $r_{ij}$ will always be positive which allows us to avoid degenerate cases.

If the model $M_{i,j}$ were the Bayesian classifier then the so-called Bradley-Terry equation would hold (Hastie & Tibshirani, 1998):

$$r_{ij} = \frac{p_i}{p_i + p_j}, \tag{2}$$

which we can rewrite as

$$p_j = \frac{1 - r_{ij}}{r_{ij}} p_i \tag{3}$$

The last equation shows that if any $p_i$ is known, then we can deduce the value of any other $p_j$ for $j$'s that are connected to $i$ by a path in $E$.

If $E$ forms a spanning tree then, then all probabilities $p_i$ are uniquely determined by the total probability requirement

$$\sum_{i=1}^{C} p_i = 1.$$

The case when $E$ forms a spanning tree is the *minimal* subset of SVM models required to deduce multi-class probability distribution $(p_i)$.

It is more common to consider the *maximal* case when $E$ corresponds to all edges of the complete graph on $C$ vertices. Since pairwise models $M_{i,j}$ will be only approximations to Bayesian classifiers, one may expect that Bradley-Terry equations will not be consistent and thus the complete set of Bradley-Terry equations will be over-determined.

There are multiple ways to solve the equations, but the most common is (the second) method of Wu-Lin-Weng (Wu et al., 2004), which we abbreviate as WLW2. It estimates $\mathbf{p}$ by minimizing the quadratic form

$$\min_{\mathbf{p}} \frac{1}{2} \sum_{i=1}^{C} \sum_{j:j\neq i} (r_{ji} p_i - r_{ij} p_j)^2.$$

## 3 Comparisons of methods to construct SVM models

In this section, we present the results of targeted experiments that compare different methods to construct SVM models to discern the optimal one. We start by confirming that there is indeed a performance gap between the minimal and maximal models and proceed with experiments that suggest answers to the three questions Q1–Q3 from the introduction: which model graph to start with, how to combine predictions in a non-complete model graph, and which edges to add to a graph. We will use 3–6 class subsets of ten class subsets CIFAR-10, Imagenette, and Imagewoof, mainly because using smaller datasets allows to repeat experiments. All results reported in this section are averaged over 20 different networks.

### 3.1 The performance gap between minimal and maximal models

We start by comparing minimal and maximal models on three class problems. Our hypothesis is that maximal models perform better than minimal ones.

Suppose there are classes $C_1, C_2$, and $C_3$ that we want to classify. We can train three pairwise SVM models: $M_{12}$ discriminating $C_1$ from $C_2$, and similarly $M_{13}$ and $M_{23}$. From these we can construct three minimal models: the first $MM_1$ by combining $\{M_{12}, M_{13}\}$, the second $MM_2$ by combining $\{M_{12}, M_{23}\}$, and the third $MM_3$ by combining $\{M_{13}, M_{23}\}$. Finally, we can construct the maximal model by using the WLW2 coupling method which combines predictions of $M_{12}, M_{13}$, and $M_{23}$. Denote the test error rate of model $MM_i$ by $m_i$ and the test error rate of the maximal model by $w$. Finally, set

$$m = \frac{m_1 + m_2 + m_3}{3} \tag{4}$$

$$b = \min(m_1, m_2, m_3) \tag{5}$$

We evaluated mean values and standard errors of $m, w$, and $b$ for each ten class dataset when $C_1, C_2$, and $C_3$ varied over all $\binom{10}{3}$ possible triples of classes. The results are shown in Table 3.

| Dataset | Kernel | $m$ | $w$ | $b$ |
|---------|--------|-----|-----|-----|
| cifar10 | linear | 2.1 (1.5) | 1.9 (1.4) | 1.9 (1.4) |
| cifar10 | radial | 2.0 (1.4) | 2.0 (1.4) | 2.0 (1.4) |
| imagenette | linear | 1.0 (0.8) | 0.6 (0.6) | 0.5 (0.6) |
| imagenette | radial | 0.6 (0.6) | 0.6 (0.6) | 0.5 (0.5) |
| imagewoof | linear | 3.2 (3.5) | 2.5 (3.3) | 2.3 (3.3) |
| imagewoof | radial | 2.7 (3.2) | 2.5 (3.2) | 2.3 (3.1) |

Table 3: Comparison of test errors for average minimal, maximal models, and the best minimal models trained on triplets of classes. The mean values are reported before parentheses, the standard errors are inside parentheses.

The results confirm our hypothesis, since the maximal model outperforms the mean minimal models four out of six times, with the other two cases being tied (both for radial kernel). Somewhat surprisingly, the best minimal model is better that the maximal model four out of six times, with the other two cases (cifar-10) being tied. This indicates that there are cases when the maximal model may be inferior not only due to the expensive quadratic complexity with respect to the number of classes but also in terms of the accuracy of the resulting classifier.

### 3.2 Selecting the best initial minimal model

We will now investigate the answer to question Q1, namely what is the best minimal model to start building larger ones.

The model graphs of minimal models correspond to spanning trees on the set of vertices. With a growing number of classes the number of spanning trees grows quickly. In fact, for $C$ classes there are $C^{C-2}$ different

trees. It is infeasible to try every out of them even for moderate values of $C$, even though the final result of the previous section indicated it may yield performance superior to the maximal model.

Our intuition is that deduction of likelihoods along longer paths using (3) compounds errors, and therefore it would be better to have a spanning tree where paths are as short as possible. Based on this intuition we formulate the hypothesis that it is best to use a spanning tree of diameter two. To verify this hypothesis we consider five class classification problems. For five classes there are three isomorphism classes of spanning trees as shown in Figure 1.

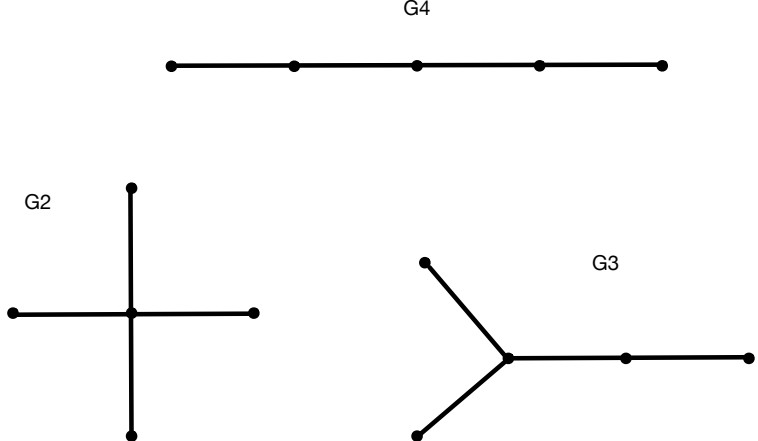

Figure 1: Trees on five vertices

In Table 4 we report on average errors made by the various models for 200 randomly selected 5-class subsets of ten classes datasets.

| Dataset | Kernel | G4 | G3 | G2 |
|---------|--------|-----------|-----------|-----------|
| cifar10 | linear | 4.7 (2.4) | 4.4 (2.1) | 4.1 (1.6) |
| cifar10 | radial | 3.6 (1.4) | 3.6 (1.3) | 3.5 (1.3) |
| imagenette | linear | 2.9 (2.2) | 2.5 (2.0) | 2.2 (1.6) |
| imagenette | radial | 1.2 (0.7) | 1.2 (0.7) | 1.1 (0.7) |
| imagewoof | linear | 7.4 (4.1) | 6.9 (3.9) | 6.4 (3.8) |
| imagewoof | radial | 5.2 (3.2) | 5.1 (3.2) | 5.0 (3.1) |

Table 4: Comparison of error rates of minimal and maximal models on five class subsets. Mean values are reported before parentheses, standard errors inside parentheses.

Among the minimal models, the one corresponding to a star on five vertices (G2) is the most accurate in all six cases. Graph $G3$ was more accurate than $G4$ in four out of six cases, with the remaining two (both for radial kernel) being tied.

This experiment indicates that it is advantageous to use a spanning tree of diameter two. There is up to isomorphism only one such tree for any value of $C$, namely the star. Therefore we propose to use the star graph (with a randomly chosen center) as the starting point for building incremental SVM models.

### 3.3 How to make predictions for non-complete model graphs

Now we turn to the second question from the introduction, namely suppose we are given a non-complete set of SVM trained in a one-vs-one fashion, how to build from them a classification model. Our solution is the most direct one, namely, manufacturing pairwise likelihood for the missing pairs and using a standard WLW2 coupling method to deduce predicted distribution over the classes. The key insight is that if we have

an estimate $o_{ik}$ of $p_i/p_k$ and an estimate $o_{kj}$ of $p_k/p_j$ then we can estimate

$$\frac{p_i}{p_j} = \frac{p_i/p_k}{p_k/p_j} \approx \frac{o_{ik}}{o_{kj}}. \tag{6}$$

Now if, we start building an incremental model with a star centered at $K$, then when a model discriminating class $i$ and $j$ is missing we can always use (6) with $k = K$ to get an estimate of $p_i/p_j$. There may be however other values of $k$ for which both $p_i/p_k$ and $p_k/p_j$ are known from binary SVM models. The question is how to combine them. We provide evidence that simple averaging is sufficient with two experiments.

In the first experiment, we select four classes $C_1, C_2, C_3$, and $C_4$ from a ten-class dataset and use four pairwise SVM: the one discriminating $C_1$ from $C_2$, the one discriminating $C_2$ from $C_3$, the one discriminating $C_3$ from $C_4$ and the one discriminating $C_4$ from $C_1$. Then we compute the correlation of estimates of $\log(p_1/p_3)$ using $k = 2$ and $k = 4$, as well as the correlation of estimates of $\log(p_2/p_4)$ using $k = 1$ and $k = 3$. We repeat this 200 times for each dataset and kernel. The results are shown in Table 5.

| Dataset | Kernel | Mean correlation | Standard deviation |
|---------|--------|------------------|--------------------|
| cifar10 | linear | 0.986 | 0.009 |
| cifar10 | radial | 0.992 | 0.004 |
| imagenette | linear | 0.973 | 0.017 |
| imagenette | radial | 0.994 | 0.005 |
| imagewoof | linear | 0.963 | 0.029 |
| imagewoof | radial | 0.992 | 0.005 |

Table 5: Correlation between estimates of log odds of pairwise likelihoods using different $k$ in (6).

We can see that estimates are highly correlated, which means fitting even a simple linear model would be unstable. This suggests using the average of log odds to combine predictions. Another alternative, possibly more robust, would be the median, and we evaluate it in our second experiment. For this experiment, we choose a six-class subset of ten class dataset and assume the model graph is the complete bipartite graph $K_{3,3}$. For each missing SVM, we can estimate it using three different choices of $k$. We repeat it 200 times for each dataset and kernel. The results are shown in Table 6.

| Dataset | Kernel | median | mean |
|---------|--------|--------|------|
| cifar10 | linear | 4.3 (1.3) | 4.2 (1.3) |
| cifar10 | radial | 4.1 (1.2) | 4.1 (1.2) |
| imagenette | linear | 1.5 (0.6) | 1.4 (0.6) |
| imagenette | radial | 1.2 (0.5) | 1.2 (0.5) |
| imagewoof | linear | 5.9 (3.1) | 5.8 (3.0) |
| imagewoof | radial | 5.6 (2.8) | 5.6 (2.8) |

Table 6: Comparison of error rates of models on six class subsets of ten class datasets when mean versus median are used for aggregation of estimates using (6). Mean values are reported before parentheses, standard errors inside parentheses.

We can see that the results are very close, with the mean edging the median by a minimal margin in the case of the linear kernel. We conclude that averaging of predictions appears to be a suitable way of aggregating multiple estimates using (6). Explicitly we will use the estimate

$$\frac{p_i}{p_j} \approx \left( \prod_{k \in O_{ij}} \frac{o_{ik}}{o_{kj}} \right)^{1/|O_{ij}|}, \tag{7}$$

where $O_{ij}$ is the set of classes $k$ for which estimates of both $p_i/p_k$ and $p_k/p_j$ are known from a pairwise model.

### 3.4 Choosing which model to incrementally add

In this section we tackle the third question Q3 posed in the introduction, that is which models to incrementally add to maximize accuracy. Our intuition is that it is most valuable to add the SVM corresponding to the hardest unsolved one-vs-one classification problem. We conducted an experiment on four-class subsets of ten class datasets aiming to confirm this hypothesis.

We started with a star graph on four vertices which has 3 edges. We selected 200 samples from the training set for each class. We computed the confusion matrix $X$ corresponding to the star graph for the chosen samples and let $Y = X + X'$ be the symmetrized confusion matrix. Then we measured accuracy after adding each of the remaining three edges in the complete graph on 4 vertices. We named the three possible models $A1, A2$, and $A3$, with $A1$ corresponding to adding an edge having the smallest entry in $Y$, while $A3$ having the largest entry in $Y$. We did this for all possible arrangements and computed averages and error rates which are shown in Table 7. In five out of six cases, model $A3$ turned out to be best, with the remaining case tied. We conclude that our hypothesis turned out to be true in these cases.

| Dataset | Kernel | A1 | A2 | A3 |
|---------|--------|----|----|----|
| cifar10 | linear | 3.1 (1.6) | 3.0 (1.6) | 2.8 (1.4) |
| cifar10 | radial | 2.8 (1.4) | 2.8 (1.4) | 2.8 (1.4) |
| imagenette | linear | 1.6 (1.3) | 1.4 (1.2) | 1.0 (0.7) |
| imagenette | radial | 0.9 (0.7) | 0.9 (0.6) | 0.8 (0.6) |
| imagewoof | linear | 4.8 (3.7) | 4.6 (3.7) | 3.8 (3.3) |
| imagewoof | radial | 3.9 (3.2) | 3.8 (3.2) | 3.6 (3.2) |

Table 7: Comparison of error rates of intermediate SVM model depending on which edge was added to the minimal one. Mean values are reported before parentheses, standard errors are inside parentheses.

### 3.5 Proposed algorithm

Let us explicate our proposed algorithm based on the experiments in the previous section.

---

**Algorithm 1** Incremental multi-class SVM model creation

---

1: **procedure** GROW_SVM($T$, $V$)
2:    # Argument $T$ is a training dataset with $N$ classes
3:    # Argument $V$ is a validation dataset (possibly $V \subset T$)
4:
5:    Choose a spanning tree with star topology at random. Denote by $E$ the set of its edges.
6:    **for** $e := (i, j) \in E$ **do**
7:        Train the probabilistic pairwise SVM model on $S$ distinguishing classes $i$ and $j$
8:    Set $step \leftarrow 1$
9:    Set $F$ to be the complement of $E$ in the set of edges of the complete graph on $N$ vertices.
10:   **while** $F$ is nonempty **do**
11:       **yield**($tuple$(step = $i$, model graph = $E$))
12:       Compute confusion matrix $X$ on $V$ using (7) to fill missing pairwise odds
13:       Set $Y = X + X'$
14:       Choose an edge $f = (i, j)$ in $F$ such $Y_{ij} = \max_{(m,n) \in F} Y_{mn}$
15:       Train probabilistic pairwise SVM model on $S$ distinguishing classes $i$ and $j$
16:       Set $F \leftarrow F - \{f\}$
17:       Set $E \leftarrow E \cup \{f\}$

---

One possible concern about the algorithm is the need for a validation dataset. In the previous section, we used a subset of the train set. We devised an experiment to verify that the choice does not have a significant impact on the accuracy of the model. For the imagenette dataset, we trained 20 networks, but

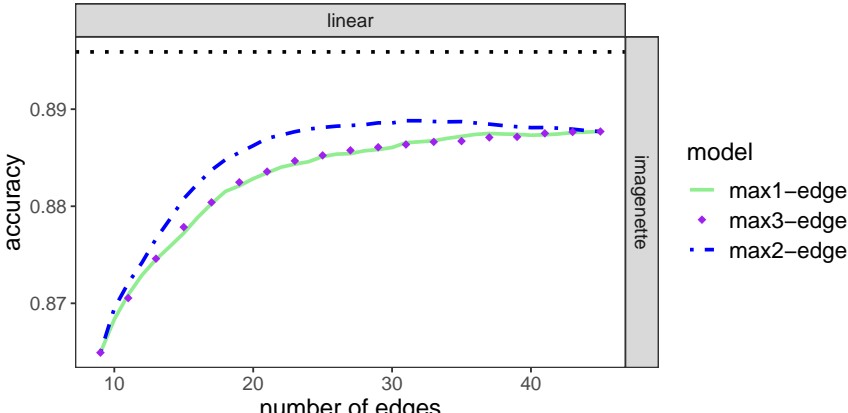

Figure 2: Comparison of mean accuracy of GROW_SVM algorithm depending on the choice of validation dataset. The dotted line indicates the accuracy of the neural network.

before training, we set aside 200 extra samples for each class from the train dataset. Then we compared three different choices of validation dataset $V$:

- *max1-edge* uses for $V$ samples data from the training dataset without extra samples, namely 200 samples per class,

- *max2-edge* uses for $V$ the test dataset,

- *max3-edge* uses for $V$ used the extra samples omitted from the training dataset.

Figure 2 shows that the accuracy of the first and the third algorithm is very close. This bodes well for applications, which often cannot set aside a special validation dataset. Although algorithm *max2-edge* performed best, one should note that this is not a valid choice, because using a test dataset in model building represents a form of test leakage.

## 4 Evaluation of intermediate SVM models

In this section, we will evaluate the performance of intermediate models based on accuracy and robustness. Accuracy evaluation will investigate the following questions:

- how does an intermediate model compare to the accuracy of the neural network,

- how does the expected accuracy increase (or possibly decrease) with the increasing number of edges,

- how does an the accuracy of *max1-edge* model compare to the alternatives.

For the third point, we considered the following alternatives:

- **random-edge** We start with a random star graph, and at each step add a random edge.

- **random-star** We start with a random star graph and at each step add a randomly chosen star.

- **vertex-transitive graphs** of diameter 2: complete bipartite graph $K_{n,n}$ on $2n$ vertices, Petersen graph on 10 vertices, and Hoffman-Singleton graph on 50 vertices (Hoffman & Singleton, 1960).

All of these alternatives yield model graphs of diameter 2, which means we can apply (7) to estimate missing pairwise likelihoods. Compared to the max1-edge algorithm, these alternatives do not require a validation dataset, although the resulting speed-up is inconsequential.

For ten-class datasets, we averaged over 20 different networks and ten possible centers of the initial star. For Imagenet-50 we used a single network, but tried all 50 possible centers of the initial star.

## 4.1 Discussion

The plots of average accuracy versus the number of edges in the model graph are shown in Figure 3.

Neural networks are more accurate than SVM except for the imagewoof dataset. For Imagenet-50, the performance of neural networks and SVM models is very similar.

Dependence on the number of edges in the experiments turned out monotonic, meaning maximal models showed the best performance. For 10 class datasets, most of the accuracy was achieved by including about half of all possible edges. For Imagenet-50 only about 250 edges sufficed to capture most of the accuracy of the maximal model, which is about 20% of the total.

Compared to alternative intermediate models, max1-edge outranked the rest. The second best was the random-star algorithm, but with the increasing number of edges its performance suffered and approached that of random-edge algorithm. Among vertex transitive graphs only $K_{5,5}$ showed performance above the random-edge algorithm, but still well below max1-edge.

For the radial kernel, the accuracy gap between minimal and maximal models is much smaller compared to the linear kernel mirroring results in section 3.1.

We have plotted the standard deviation of predictions on Imagenet-50 for the various methods in figure 4. The key point is that max1-edge was again the best, and the error stabilized with $\sim 250$ edges, which is about the same number of edges as when its accuracy reached the plateau. Remarkably, the Hoffman-Singleton graph showed very stable behavior on par with max1-edge. This is surprising since its accuracy was lagging.

## 5 Conclusion

We exhibited a variety of intermediate multi-class SVM models trading accuracy for training time complexity. Our experiments uncovered key properties improving their accuracy. The max1-edge variant of the proposed GROW_SVM algorithm showed superior accuracy across diverse datasets. It achieved near-optimal accuracy while requiring only 20-50% of the computational cost of maximal models. This improvement is meaningful for large values of $C$, when model construction can take hours or days. The method crucially uses a confusion matrix in the training phase, rather than relegating it to a mere tool for model evaluation. The algorithm is quite general; it is neither restricted to problems arising from neural networks nor to a specific kernel. The efficacy of the method will vary according to a dataset, but the incremental approach allows for monitoring of model-building progress and early stopping when incremental accuracy gains stall.

Our investigation reaffirms the efficacy of established methods. Softmax outperformed SVM models except for the challenging imagewoof dataset. This shows the benefit of whole model optimization employed in training neural networks which contrasts the bottom-up approach of SVM. We also confirmed that commonly used maximal SVM models are very accurate. In our experiments, they were on average more accurate than models with smaller model graphs. Finally, the radial kernel proved surprisingly versatile, since it achieved near optimal accuracy on tasks, for which interclass boundaries were expected to be very close to linear.

Our work underscores the crucial choice of model graph for intermediate probabilistic SVM. One may hypothesize that special classes of graphs, such as Cayley graphs or expander graphs (Hoory et al., 2006), could be useful for building multi-class SVM models. For that to happen, we would need a new way to conduct multi-class inference using intermediate graphs, unlike the one we used in the paper, which is limited to graphs of diameter two.

## 6 Acknowledgments

The first author was supported by VEGA grant 2/0172/22 "Classification using ensembles of neural networks" and by the Operational Programme Integrated Infrastructure (OPII) for the project 313011BWH2: "InoCHF

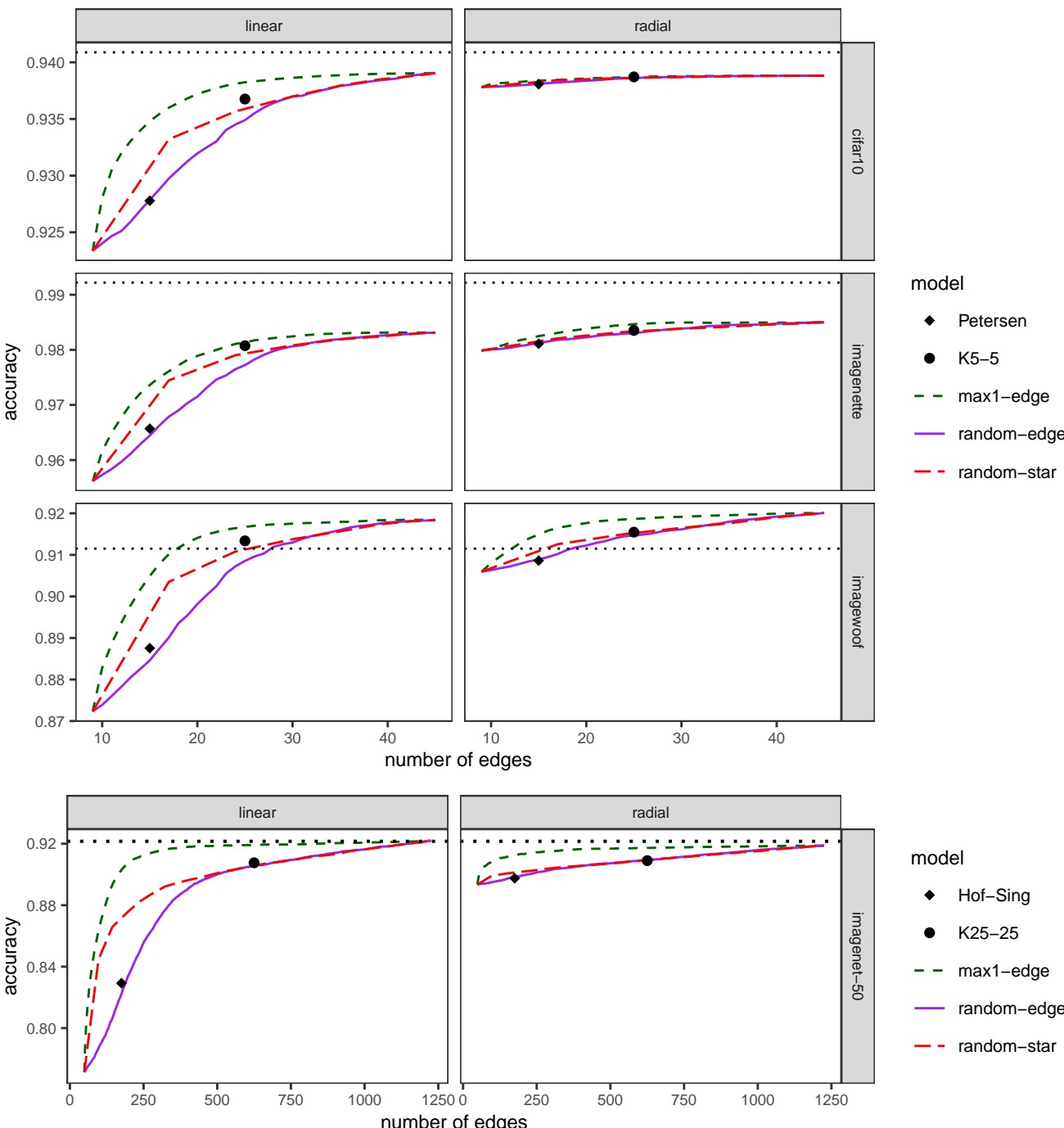

Figure 3: Average accuracy plotted against the size of the model graph for Imagenet-50. Hof-Sing denotes results for the Hoffman-Singleton graph, K5-5 stands for the complete bipartite graph $K_{5,5}$ and K25-25 stands for $K_{25,25}$. The dotted horizontal line indicates the mean accuracy of neural networks.

– Research and development in the field of innovative technologies in the management of patients with CHF", co-financed by the European Regional Development Fund.

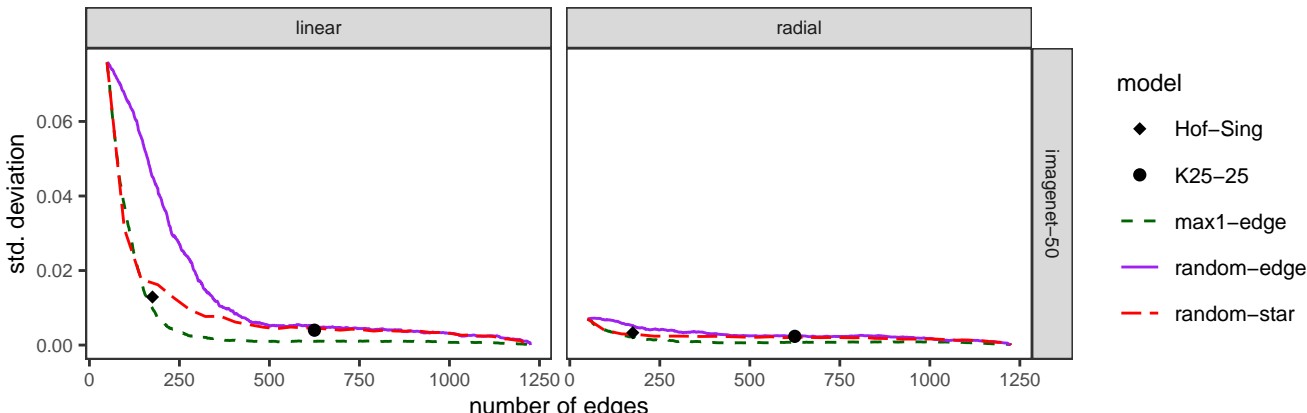

Figure 4: Mean standard deviation of the accuracy of intermediate models plotted against the size of the model graph for Imagenet-50. Hof-Sing denotes results for Hoffman-Singleton graph, and K25-25 stands for $K_{25,25}$.

The second author was supported by Operational Program "Integrated Infrastructure" of the project "Integrated strategy in the development of personalized medicine of selected malignant tumor diseases and its impact on life quality", ITMS code: 313011V446, co-financed by resources of European Regional Development Fund.

Authors would like to thank anonymous reviewers for their valuable suggestions.

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
