# OpenReview forum: "Bridging performance gap between minimal and maximal SVM models"
_TMLR — Accepted by TMLR_

### Review · Reviewer_3QF9 · 2022-10-23

**Summary Of Contributions:**

Support vector machine (SVM) is a commonly used traditional machine learning method for effectively handling binary classification. SVM itself does not support multi-class classification and requires meta-strategies, which may lead to a large increase in computation. An interesting question is whether there is a model that requires less training time but still maintains good accuracy. In this paper, the author(s) opened an interesting exploration of this from the perspective of metrical and topological properties, and proposed two ways to build smaller multi-class SVM models.



**Audience:**

Yes

**Broader Impact Concerns:**

No.

**Claims And Evidence:**

No

**Requested Changes:**

Detailedly, I have the following comments/questions:

1. For the datasets used in this paper, it is suggested to give a table with headings such as the name of datasets, number of samples, number of features, and so on. This will make it clear to the reader(s). And for Table 2 in the paper, please give the titles of columns; also, the caption width setting seems too narrow.

2. On Page 4, Section 2.3, Paragraph 2, Lines 3-4, the author(s) mentioned that "where we divide the training data into four folds and train four SVM models", so, is 4-fold cross-validation used? Then for the results in the paper, for example, in Table 3, is it averaged over  $\tbinom{10}{3}$ * 4 calculations?

3. For the results in the paper (for example, Tables 3-5), it is better to give the average with the standard error. Since many results in the paper are actually close in terms of the average, maybe the fluctuations of the models can be better observed by using standard errors.

4. In Tables 3-5, why are the error rates opposite on datasets imagenette and imagewoof for different methods whole and fresh? Could you please explain it? In addition, in Tables 3 and 4, maybe the accuracy results of neural networks can also be given as a reference.

5. On page 8, Line 3 below Table 5, the author(s) mentioned that "the neural network is more accurate than the maximal models", I think that their results appear to be close (?).

6. In Figure 3, how to calculate the distances between classes.

7. On Page 1, Section 1, the last sentence in the 2nd Paragraph, the author(s) mentioned that "...have drawbacks e.g. how to resolve tied votes...", is this really a drawback of voting ensembles? I think it might be easily fixed by requiring an odd number of people/models to vote.

8. Some other tiny issues/typos:

(1) Regarding the word "SVM" in the Abstract, when the abbreviation appears for the first time, please give its full name. Similar to the first sentence in the Introduction.

(2) In Abstract, Line 4, "metricàl" should be "metrical"?

(3) On Page 3, Section 2.2, Line 2, what does "(Page)" mean?

(4) On Page 4, Section 2.3, Paragraph 2, Lines 2-3, "testing test" should be "test set"; the word "inthe" should be "in the"?

(5) In Tables 3-5, maybe there is no need to repeat the names of datasets and models so many times.

(6) In Figures 7 and 9, some of the text on the picture is obscured

In addition, the format of the references is quite inconsistent. Please check carefully and correct it.

I look forward to hearing from the author(s). Also, there is not a lot of overlap between my research and the work of this paper, please correct me if I am wrong. Thanks.





**Strengths And Weaknesses:**

## Strong points:

1) (Trying to) improve the SVM multi-class classification model effectively and efficiently.

2) Empirical experiments are performed to validate the discussion in this paper.

---

## Weak points:

Although the work of this paper looks interesting, there are some points in this paper that are not so clear to me, I hope the author(s) could explain/detail them. Thanks.


1) The technical details of empirical experiments are not described clearly, for example, how to calculate the averaged errors; the analysis of the empirical results is not thorough, for instance, why the error rates are opposite on datasets imagenette and imagewoof for different methods whole and fresh (Please see Section "Requested Changes" below for details).

2) There are some issues that need to be clarified, such as how to calculate the distances between classes (Please see Section "Requested Changes" below for details).

In addition, this paper has some empirical experiments, but I am not sure the empirical description/experiments would be enough to reproduce because no code of this paper seems to be provided.

---

---

> ### Author Response · Authors · 2022-10-29
> **Revision of paper in progress**
>
> Thank you for helpful comments, we are incorporating changes to reflect your questions and suggestions. We were not aware that it is possible to publish an anonymized code repository. We are in process of building one so that anyone can replicate our experiments.

---

> > ### Author Response · Authors · 2022-11-12
> > **Repository created**
> >
> > We have created an anonymized repository with the code used to carry out experiments. It can be found at
> >
> > https://anonymous.4open.science/r/intermediate-svm-FAE3/README.md
> >
> > We have added a graph showing boxplot of experiments of section 3.3, and a graph showing standard deviations of experiments in Section 4.4. They can be found in the directory 'plots' as files standard-errors1.pdf and standard-errors2.pdf.
> >
> > Please note that not all intermediate files were added due to their size.

---

> > ### Comment · Action_Editors · 2022-12-10
> > **status of the revision**
> >
> > Dear Authors,
> >
> > Have you submitted the revised version of the paper? Maybe I missed something from your message.
> > Thanks
> >
> > AE

---

> ### Author Response · Authors · 2022-11-28
> **Large intermediate files added to code repository**
>
> We took advantage of git-lfs extension and added larger intermediate files to the code repository. The new version with larger files is
>
> https://anonymous.4open.science/r/intermediate-svm-3810/README.md
>
> The paper is still undergoing rewrite. Some experiments are going to be removed as they do not add essential value, while others are added to fill logic gaps and enable comparison with other state-of-the-art methods.

---

### Review · Reviewer_Gmnt · 2022-11-01

**Summary Of Contributions:**

This paper reports the results of experimental comparisons of a few aggregation methods for multiclass SVM that aggregates several one-vs-one SVMs.
The experiments are conducted on some image datasets where images are transformed to feature vectors using CNNs.
The authors reported that aggregations of one-vs-one SVMs for all class pairs (called *maximal*) attained the smallest classification errors.
The authors then considered graphs representing the pairwise relationships between the classes.
The authors then reported that, by adding edges to the graph randomly or by some strategies (i.e., selecting one-vs-one SVM to be added for aggregation), we can obtain aggregated multiclass SVM with almost the same accuracy as the maximal aggregation when we added around the half of the all possible pairs.

**Audience:**

No

**Broader Impact Concerns:**

There is no ethical concern.

**Claims And Evidence:**

No

**Requested Changes:**

**Request 1.**
Please describe the detailed background, research question, and the contribution of this research.
Why and how the research topics under considerations are important?
What is the research question addressed in this study?
What is the answer to the research question?

**Request 2.**
Please describe the appropriateness of the experimental setups with respect to the underlying research questions.
What are the questions addressed in each experiment?
Are the choice of the experimental setups (such as the choice of the datasets) appropriate to answer the questions of interest?
For what purposes or in which contexts, the experimental setups are relevant?

**Request 3.**
Please review related studies more exhaustively.
It is highly questionable if there are only a few studies conducting similar comparisons as in the current paper.
Is the finding reported in the paper truly novel?
Are there anyone reported similar findings before?

**Strengths And Weaknesses:**

### Strong aspects

I could not find any strong aspects.


### Weak aspects

This paper has several fatal weaknesses, as follows.

**Weakness1: The research questions as well as the contributions are not clear.**

Introduction merely states that "The primary aim of this work is to study the performance implications of the choice of model’s graph.", without explaining why and how this is an important research topic.
This lack of research background is fatal for evaluating the contribution of the paper.
The lack also makes difficult to assess the appropriateness of the experimental setups (see Weakenss2 below as well).

**Weakness2: The appropriateness of the experimental setups are not described.**

Most parts of the paper focus on the detailed procedures of the experiments.
However, the purposes of each experiment are not described in the paper.
For example, in Section 3, the authors compared minimal and maximal SMVs.
There, the authors merely stated that "We study the performance gap between minimal and maximal SVM models, showing that there is a measurable gap."
From the no free lunch theorem, we know that no single algorithm can dominate the others.
Hence, the comparisons of the algorithms cannot be independent from the choice of the datasets.
Although the authors insisted that their choice of the datasets has "several compelling attributes" in page 2, there are no discussions on how the choice of the datasets are relevant to the comparisons under consideration.
Without such a discussion, what implications should we find from the comparisons on these datasets?
Apparently, the results on these datasets do not guarantee that the findings reported in the paper to hold on *any* datasets.
For what purposes or in which contexts, the comparisons on these datasets are relevant?
The same questions also arise in Section 4 as well.

**Weakness3: The related studies are not reviewed appropriately.**

Only a few related studies are mentioned in the paper; the authors contrasted the current study with LIBSVM in Table 1, and the resemblance of the found results to (Šuch et al. (2015)) and (Šuch & Barreda (2016)) are pointed out in Section 5.5.
It is highly questionable if they are the only studies conducting similar comparisons as in the current paper.
This question leads to fundamental concerns.
Is the finding reported in the paper truly novel?
Are there anyone reported similar findings before?

---

### Review · Reviewer_ev2n · 2022-12-10

**Summary Of Contributions:**

This paper studied the one-versus-one classification strategy used in multi-class SVM. The author(s) proposed two methods to build the one-versus-one comparison graph. Experiments on Image classification tasks are conducted to support the effectiveness of the proposed method.

**Audience:**

Yes

**Broader Impact Concerns:**

I do not have any concerns on the ethical implications of this work.

**Claims And Evidence:**

Yes

**Requested Changes:**

- Add a related work section and clearly describe the position of this work in the literature.
- Add some discussion on the one-versus-all method and compare it with one-versus-one.
- Add some analysis on the time complexity of the proposed method.
- Add some new experiments with some datasets from the extreme classification repo.

**Strengths And Weaknesses:**

The paper is well-written and easy to follow in general, but I still have some concerns on the current manuscript.

Some concerns:

- It seems that the related work section is missing. I encourage the author(s) to carefully describe some related works so readers can better understand the position of this work in the literature.
- I understand that the focus of this paper is one-versus-one strategy, but I think it is still necessary to include some discussion on one-versus-all method such as softmax, e.g., when should we use one-versus-one and when one-versus-all is preferred?
- It is better to include the time complexity of the proposed method (the time complexity to build the graph and the time complexity of solving those one-versus-one subproblems). I also suggest the author to compare the time complexity to the softmax method.
- The number of classes is actually not large for the experimental datasets. I encourage the author(s) to include some datasets from the extreme classification repo (http://manikvarma.org/downloads/XC/XMLRepository.html). These datasets have a larger number of classes and seem to be suitable for the evaluation of the proposed method.

Minor concerns:
- page 4, inthe work -> in the work
- page 4, then, then -> then

---

### Decision · Action_Editors · 2023-02-26

**Recommendation:** Accept as is

**Comment:**

This paper studied the one-versus-one classification strategy used in multi-class SVM. Two new methods were proposed to build the one-versus-one comparison graph. Experiments on image classification tasks are conducted to validate the effectiveness of the proposed method.  Two reviewers recommended acceptance since they agreed that the revised version has addressed their concerns.  Reviewer ev2n recommended "rejection" because the reviewer thought that the authors did not provide the revised version. However, the authors did provide a revised version which was delayed for a couple of weeks.  The revised version did address Reviewer ev2n's main concern in Section 1.1 on the complexity analysis of the methods which I found is sufficient.

Based on the above reasons, I recommend its acceptance but not Featured Certification.

**Audience:**

Yes.

**Claims And Evidence:**

Yes.